# Testicular Caspase-3 and β-Catenin Regulators Predicted via Comparative Metabolomics and Docking Studies

**DOI:** 10.3390/metabo10010031

**Published:** 2020-01-11

**Authors:** Mohammed S. Hifnawy, Mahmoud A. Aboseada, Hossam M. Hassan, Asmaa M. AboulMagd, Adel F. Tohamy, Samraa H. Abdel-Kawi, Mostafa E. Rateb, El Moataz Bellah El Naggar, Miaomiao Liu, Ronald J. Quinn, Hani A. Alhadrami, Usama Ramadan Abdelmohsen

**Affiliations:** 1Department of Pharmacognosy, Faculty of Pharmacy, Cairo University, Cairo 11865, Egypt; mohamed.hifnawy@pharma.cu.edu.eg; 2Department of Pharmacognosy, Faculty of Pharmacy, Nahda University, Beni-Suef 62513, Egypt; mahmoud.moawwad@nub.edu.eg; 3Department of Pharmacognosy, Faculty of Pharmacy, Beni-Suef University, Beni-Suef 62513, Egypt; abuh20050@yahoo.com (H.M.H.); mostafa.rateb@uws.ac.uk (M.E.R.); 4Department of Pharmaceutical Chemistry, Faculty of Pharmacy, Nahda University, Beni-Suef 62513, Egypt; asmaa.aboulmaged@nub.edu.eg; 5Department of Toxicology and Forensic Medicine, Faculty of Veterinary Medicine, Cairo University, Cairo 11865, Egypt; adeltohamy@cu.edu.eg; 6Department of Medical Histology and Cell Biology, Faculty of Medicine, Beni-Suef University, Beni-Suef 62513, Egypt; samraa.hussein@nub.edu.eg; 7Department of Basic Science, Faculty of Dentistry, Nahda University, Beni-Suef 62513, Egypt; 8Marine Biodiscovery Centre, University of Aberdeen, Aberdeen AB24 3UE, UK; 9School of Computing, Engineering and Physical Sciences, University of West Scotland, Paisley PA1 2BE, UK; 10Department of Pharmacognosy, Faculty of Pharmacy, Damanhur University, Elbehira 22511, Egypt; moataz.elnagar@pharm.dmu.edu.eg; 11Griffith Institute for Drug Discovery, Griffith University, Brisbane, QLD 4111, Australia; miaomiao.liu@griffith.edu.au (M.L.); r.quinn@griffith.edu.au (R.J.Q.); 12Faculty of Applied Medical Sciences, Department of Medical Laboratory Technology, King Abdulaziz University, P. O. Box 80402, Jeddah 21589, Saudi Arabi; 13King Fahd Medical Research Centre, King Abdulaziz University, P. O. Box 80402, Jeddah 21589, Saudi Arabia; 14Department of Pharmacognosy, Faculty of Pharmacy, Minia University, Minia 61519, Egypt; 15Department of Pharmacognosy, Faculty of Pharmacy, Deraya University, Universities Zone, New Minia City 61111, Egypt

**Keywords:** testicular spermatogenesis, β-catenin, caspase-3, *Albizzia lebbeck*, *Anagallis arvensis*, *Rosmarinus officinalis*, metabolomic profiling, docking

## Abstract

Many routes have been explored to search for effective, safe, and affordable alternatives to hazardous female contraceptives. Herbal extracts and their secondary metabolites are some of the interesting research areas to address this growing issue. This study aims to investigate the effects of ten different plant extracts on testicular spermatogenesis. The correlation between the chemical profile of these extracts and their in vivo effect on male reproductive system was evaluated using various techniques. Approximately 10% of LD_50_ of hydro-methanolic extracts were orally administrated to rats for 60 days. Semen parameters, sexual organ weights, and serum levels of male sex hormones in addition to testes histopathology, were evaluated. Moreover, metabolomic analysis using (LC-HRESIMS), multivariate analysis (PCA), immunohistochemistry (caspase-3 and β-catenin), and a docking study were performed. Results indicated that three plant extracts significantly decreased epididymal sperm density and motility. Moreover, their effects on testicular cells were also assured by histopathological evaluations. Metabolomic profiling of the bioactive plant extracts showed the presence of diverse phytochemicals, mostly oleanane saponins, phenolic diterpenes, and lupane triterpenes. A docking study on caspase-3 enzyme showed that oleanane saponins possessed the highest binding affinity. An immunohistochemistry assay on β-catenin and caspase-3 indicated that *Albizzia lebbeck* was the most active extract for decreasing immunoexpression of β-catenin, while *Rosmarinus officinalis* showed the highest activity for increasing immunoexpression of caspase-3. The spermatogenesis decreasing the activity of *A. lebbeck, Anagallis arvensis,* and *R. officinalis* can be mediated via up-regulation of caspase-3 and down-regulation of β-catenin existing in testis cells.

## 1. Introduction

In developing countries, populations are increasing rapidly. High population numbers have the potential to lead to numerous economic, environmental, resource, and social problems [1]. Recently, researchers have given priority to the use of medicinal plants for their contraceptive properties, with a majority of them being used as female contraceptive agents, while some affect the male reproductive system [2,3]. *Albizzia lebbeck* L. Benth (*Mimosoidae*), *Anagallis arvensis* L. (*primulaceae*), and *Calendula officinalis* L. (*Astraceae*) are annual plants containing oleanane saponins which have been linked to various biological activites affecting sperm [4,5,6]. *A. lebbeck* extracts have exhibited anti-inflammatory, antiallergic, and antidiabetic activities [7,8,9]. Additionally, the saponins found in the bark of this plant are also reported to have adverse effects on the male reproductive system [10], while its saponin-rich root and pod fractions were shown to cause spermicidal action in human semen [11]. *A. arvensis* extract has been used in folk remedies for bronchial asthma, fever, and rheumatism [12]. On top of this, anagalligenone sapogenin isolated from *A. arvensis* caused the instantaneous immobilization of spermatozoa in one minute [13]. The flower extract of *C. officinalis*, commonly known as marigold [14], has been used to treat HIV and cancer [15,16]. Moreover, the saponin-rich fraction from marigold flowers has exhibited potent spermicidal activity [17]. 

*Apium graveolens* L., *Anethum graveolens* L. (*Apiaceae*), and *Menthae piperitae* (*Lamiaceae*) are rich in polyphenolics [18,19,20] and volatile constituents [21,22,23]. Essential oil and extracts of *A*. *graveolens* (celery) have been utilized for numerous medical problems such as hepatic disorders, peptic ulcers, and hypertension [22,24,25]. It was studied for its anti-fertility effect in males, suggesting the high content of phytoestrogens (flavonoids) seemed to be responsible for decreasing sex hormone levels in the blood [26]. *Anethum graveolens* (dill) is on the vital list of plants for home complementary medicine, as its essential oil and extracts can help in GIT disorders, have antioxidant and anti-hyperlipidemic effects [27,28], and its hydro-methanolic extract adversely affects male fertility [29]. The essential oil and extracts of *M. pepperitae* (peppermint) have been used as antimicrobial and hepatoprotective agents and to improve menstrual disorders [30,31,32]. Peppermint tea has been reported to affect male sex hormones levels and its essential oil has been shown to possess potent spermicidal activity [33,34].

*Rosmarinus officinalis* L. (rosemary) is a common household plant containing mainly phenolic diterpenes, triterpenes, polyphenolics, and other essential oil constituents [35,36,37]. Extracts of rosemary have been used as antipyretic and anti-inflammatory agents and to relieve respiratory disorders [38,39]. Administration of high doses of rosemary leaf extract to male albino rats led to adverse effects on the male reproductive system [40]. 

*Hibiscus sabdariffa* L. (rosel) is also considered as one of the polyphenol-rich plants, especially high in anthocyanins [41]. Their extracts made from the calyces possess a remarkable antihypertensive effect [42]. It was reported that sub-chronic administration of calyces extract to male rats or mice caused harmful effects on testis ultrastructure, suggesting that protocatechuic acid, the main active constituent, is responsible for this effect based on its high structural similarity with polyphenolic gossypol [43,44].

*Calotropis procera Asclepiadaceae* (Ait) R.Br. is a common xerophytic perennial shrub. Aerial parts have been found to contain mainly cardiac glycosides and flavonoids [45,46]. Its root extract has been used in traditional medicine for the treatment of leprosy, ulcers, tumors, and piles [47]. Its flower extract was reported to cause functional alteration in the genital organs of male mice [47], and the crude extract of roots and flowers of *C. procera* proved to produce spermicidal action to adult male rat semen [48]. Calotropin cardenolide may be responsible for the anti-male fertility activity of *C. procera*, as reported by Sarma et al. [49].

*Lactuca sativa* (Lettuce, *Asteraceae*) was reported to contain mainly sesquiterpene lactones [50]. Its latex sap and hydroalcoholic extracts possess sedative, anticonvulsant, and antifungal properties [51,52,53]. Furthermore, hydro-alcoholic extracts of its seeds were found to exhibit antispermatogenic effects for adult male mice [54].

Dereplication has become commonplace in natural products research, allowing for the rapid identification of known metabolites in complex mixtures [55,56]. Dereplication with LC-MS and subsequent database searches, using MarinLit [57] and AntiBase [58], makes screening samples for known natural products significantly easier. It aids in saving time and reducing the possibility of redundancy of re-isolation throughout natural product discovery programs. Metabolomics can be defined as the comprehensive analysis of molecules in a biological system under a given series of conditions [59]. At the biochemical standard, the metabolome is most closely linked to the phenotype, providing insight into biological functions [60].

As part of our ongoing research on the aforementioned plants, this study aims at investigating the chemical and biological profiles of each plant. In this approach, metabolomic analysis using liquid chromatography coupled with high-resolution electrospray ionization mass spectrometry (LC-HRESIMS) was applied to initially assess and dereplicate the secondary metabolites of the tested plants. Subsequently, we evaluate datasets for correlations between their effect on the male reproductive system and the associated chemical profile in order to identify or suggest mechanistic pathways by which plant extracts influence some morphological features of male rat testes as well as semen parameters.

## 2. Results 

### 2.1. Sperm Density and Motility

The treated rats showed a significantly high reduction in the sperm concentration of cauda epididymides in extracts of *Anagallis arvensis* (AA) (*p* < 0.0001), *Albizzia lebbeck* (AL) (*p* < 0.001), and *Rosmarinus officinalis* (RO) (*p* < 0.0001). Additionally, *Hibiscus sabdariffa* (HS), *Calotropis procera* (CP), and *Anethum graveolens* (ANG) extracts exhibited a low but statistically significant (*p* < 0.05) reduction in the sperm concentration of cauda epididymides. The sperm motility of the cauda epididymis was also significantly reduced (*p* < 0.0001) in *A. arvensis*, *A. lebbeck*, and *R. officinalis* samples, and only ANG extract reduced the sperm motility, but with low significance (*p* < 0.05) (Figure 1).

### 2.2. Hormonal Assay and Effect on Organ Weights

No significant difference was realized between the weights of treated rats and their sexual organs (testis and seminal vesicle) compared to the control group. Additionally, there was no effect on sex hormones levels (testosterone, FSH, and LH) of the treated animal group in comparison to the untreated group (Appendix A). 

### 2.3. Histological Analysis

Light microscopic examination of H&E stained sections of the testis of the control adult albino rats showed densely packed seminiferous tubules forming testicular parenchyma. The tubules were lined by germinal epithelium and separated by an interstitium containing interstitial cells of Leydig and blood vessels (Figure 2A). The spermatogenic cells including spermatogonia, primary spermatocytes, spermatids, and Sertoli cells had large pale nuclei resting on the basement membrane. The tubules were enclosed by myoid cells (Figure 2B). In the RO treated group, densely packed seminiferous tubules but with interstitial exudates were observed (Figure 2C). Additionally, loss of the normal architecture of spermatogenic cells with partial separation from the basement membrane and the appearance of numerous empty spaces along with the shedding off in the lumen were also observed (Figure 2D). Distorted irregular seminiferous tubules with interstitial exudates and disorganization of spermatogenic epithelium (Figure 2E) were shown in the AL treated group. In addition, some tubules showed a marked reduction of the germinal epithelium with the appearance of numerous empty spaces (Figure 2F). Finally, after light microscopic examination of H&E stained sections of the testes of AA treated rats, seminiferous tubules were shown with marked reduction, detachment, displacement, and degeneration of spermatogenic cells (Figure 2G,H). No significant histological changes were observed in both seminal vesicles of all groups and testis specimens of the other seven groups compared to control groups (Appendix A).

### 2.4. Multivariate Analysis 

Processed data were analyzed by SIMCA-P V 13.0 (Umetrics, Umeå, Sweden) using principal component analysis (PCA), the unsupervised statistical analysis method. PCA was utilized to detect differing features found in the outlying extracts to aid the prioritization of the extracts with interesting bioactive metabolomes. Two predominant outliers, AA and AL were observed, indicating that there was high variance in the secondary metabolites found in these two extracts as they lay farthermost from the main group of aggregated samples in the score plot (Figure 3).

### 2.5. Chemical Diversity of Natural Products in Plant Extracts

In this ongoing study, the mass resolution was 50,000 (at *m*/*z* 400), that is high enough to distinguish closely related compounds. The total number of features detected in ten plant extracts by LC-HRMS is documented in the Appendix A, and the highest numbers of features were detected in the two outlaying extracts in addition to the bioactive RO extract is documented in Table 1, Figure 4.

### 2.6. Dereplication of A. arvensis, A. lebbeck, and R. officinalis Extracts

The crude hydromethanolic extracts of AA, AL, and RO were active on the male anti-fertility screen in the target-based functional assay on semen parameters and histopathological effect. Most of the metabolites from the AA and AL extracts were putatively assigned as saponins through dereplication (Appendix A). Several of those were dereplicated as triterpenoid saponins (oleanane) which have already been described from the two plants [61,62]. Other plausible congeners were detected such as flavonoids, phenolic acids, triterpenes, sterols, and heterocyclic amines (Appendix A). RO extract contained many metabolites, including flavonoids, phenolic acids, triterpenes (lupane derivatives), and phenolic diterpenes (Appendix A). By applying an algorithm to the data from this extract to calculate the intensity of each *m*/*z* of parent ion peaks, a plethora of metabolites with high intensities were detected, especially triterpenes (lupane derivatives) and phenolic diterpenes (Table 1).

### 2.7. Modelling Study

In this present work, all saponin derivatives, triterpenes (lupane) and phenolic diterpenes (abietatriene) could induce apoptosis via the activation of caspases. To confirm this speculation, docking studies were performed to position all of these compounds into the caspase-3 (1GFW) active site. This would allow for the prediction of apoptotic activity of the investigated constituents and to gain further insight into their relative binding affinities and binding interactions with the caspase-3 active site. The maximum number of poses per ligand was set to 10 and no constraints were used to perform molecular docking. The docking complex assumed to represent ligand-receptor interactions was selected based on three criteria: (i) binding affinity score of the pose, (ii) its orientation into the active site in a similar manner as the co-crystallized ligands orientation, and (iii) the preservation of binding interactions, especially hydrogen bond formation. The estimated binding affinity of the co-crystallized ligand was −11.425 Kcal/mol, with four conventional hydrogen bonds observed with residue His 121, Gly 122, Arg 207, and Met 61. Also, the indole ring interacted with residue Trp 206. This result gives some clue as to the mechanism of high feature compounds inducing apoptosis by the activation of caspase-3. Modeling studies suggest that albizeasaponin A binds to the caspase-3 binding site with a binding affinity of −15.227 Kcal/mol via the hydrogen bonding interaction of the OHs of the sugar with Met 61, Thr 62, and Cys 163 (Figure 5). 

The significant increase in the expression of caspase-3 antibodies in the testicular tissue section of the AL treated group is associated with the fact that albiziabioside A showed five hydrogen bond interactions with Met 61, Arg 207, and Cys 163 with a binding affinity of −10.731 Kcal/mol (Figure 5C,D). Anagallicin C has a binding affinity of −12.587 Kcal/mol through hydrogen bond interaction with Ser 63, Cys 163, Ser 205, and Arg 207 via 3 OHs and one oxygen atom of the sugar moiety, respectively (Figure 5E,F). Anagallosaponin II provided five hydrogen bond interaction via Cys 163, Phe 252, His 121, Arg 207 with a binding energy of −9.921 Kcal/mol (Figure 5G), which is consistent with positive caspase-3 reaction in the cytoplasm of germinal cells of the AA treated group. Anagallosaponin IX, which is the third-highest binding energy score among the investigated derivatives, showed the key binding interaction of the co-crystallized ligand which includes hydrogen bonding interactions of Arg 207 and Met 61. In addition, other interactions can be observed for the amino acids Gly 60, Thr 62, Cys 163, and Ser 205 (Figure 5H). The docking study indicated that the important interactions of the compounds in this study have resemblance to the binding mode of celastrol co-crystallized with caspase-3 enzyme via His 121, Gly 122, Cys 163, Asn 208, and Phe 252. The binding energy score is shown in Table 2.

### 2.8. Immunohistochemical Assay

#### 2.8.1. β-Catenin

The β-catenin expression in the testes of the control rats was evident in the basal part of the epithelium (Figure 6A). The intensity of β-catenin immunoexpression within the RO treated group and the AL treated group were significantly decreased (Figure 6B,C). The β-catenin expression in the testes of the AA treated group showed moderate positivity compared to the control group (Figure 6D). The mean color area percentage of β-catenin of the RO, AL, and AA groups was 2.537 ± 0.07588, 1.156 ± 0.08173, and 2.562 ± 0.06984, respectively, which were significantly decreased compared with the control group (3.506 ± 0.02716; Figure 6E).

#### 2.8.2. Caspase-3

A significant increase in the expression of caspase-3 antibodies in the testicular tissue section of both the RO treated group and the AL treated group was observed (Figure 7B,C), as compared to that of the control group (Figure 7A). The AA treated group showed a moderate positive caspase-3 reaction in the cytoplasm of germinal cells (Figure 7D). The mean color area percentage of caspase-3 of the RO, AL, and AA groups was 11.51 ± 0.5459, 6.272 ± 0.6107, and 1.85 ± 0.05124, respectively, which were significantly increased compared with the control group (0.5936 ± 0.08309; Figure 7E). 

### 2.9. Rule of Five and Veber’s Oral Bioavailability Rule of High Features Compounds

Compounds **11**–**15** detected in *R. officinalis* obeyed Lipinski’s rule of five and Veber’s oral bioavailability rule; only compounds **11**–**13** violated the logP rule with 6.64, 5.36, and 5.14, respectively. However, compounds **1**–**10** detected in *A. arvensis* and *R. lebbeck* extracts violated all the calculated properties. The existence of sugar moieties in these molecules provided more *O*-containing functionalities and hydroxyl groups, which might account for the violations (Figure 8).

## 3. Discussion 

Currently, research about male contraception as a new method for family planning worldwide has been terminated to avoid various hazards resulting from recurrent use of hormonal and non-hormonal female contraceptives [63,64]. Our study aimed to (a) confirm the effect of the investigated plant extracts on spermatogenesis and (b) search for the chemical constituents and biological mechanisms of bioactive extracts through the aid of PCA analysis, metabolomics, immunohistochemical assays, and drug modeling in order to develop herbal medicine in this scope. 

After conducting the in vivo biological activity for all ten selected extracts in this study, it was clear that *A*. *lebbeck*, *A*. *arvensis*, and *R*. *officinalis* were more potent extracts in decreasing sperm density, motility, and adverse effects on testis cells, e.g., densely packed seminiferous tubules, loss of the normal architecture of spermatogenic cells, disorganization of spermatogenic epithelium, and degeneration of spermatogenic cells (Figure 1 and Figure 2). Unsupervised multivariate analysis by PCA revealed that both AA and AL extracts were considered as strong outliers, whilst the other eight extracts would cluster together, including RO, even though it was one of the three most biologically active extracts (Figure 3). Therefore, to find an explanation for this outcome, metabolomics using LC-HRMS and dereplication of all investigated extracts were conducted to detect different metabolite classes, distinguish between them, and understand the reason for the variability in the anti-male fertility activities (Appendix A).

In accordance with metabolomics and dereplication of the ten tested extracts, it was clear that in both outliers AL and AA, oleanane saponins were detected as high features, while lupane triterpenes, phenolic diterpenes, and some polyphenolics (flavonoids and phenolic acids) were detected in high concentrations in RO (Table 1, Figure 4). Most of these metabolites were actually reported in these three bioactive plants, e.g., albizeasaponins A–B were isolated from *A. lebbeck* bark; albiziatriozide was isolated from *A. lebbeck* leaves [4,65]; anagallisins A–C, anagallosaponins II, VI, and IX were isolated from *A. arvensis* [5,62,66]; and betulinol, betulinic acid, 23-hydroxybetulinic, rosmanol, and carnosic acid were isolated from *R. officinalis* [67,68,69,70,71]. 

Extensive searches about these compound classes and their effects on spermatogenesis showed that there were numerous studies that evaluated the antispermatogenic effect of these metabolites; more specifically, the antispermatogenic effect of oleanane saponins was investigated [10,72], the effect of lupeol and lupeol-rich plant extracts on spermatogenesis was examined [73,74,75,76], and the action of diterpenes on testes was discussed [77,78]. There is no available data to confirm the adverse effects of polyphenolics on the male reproductive system except gossypol [3,79,80,81], and all studies reported about this action emphasized the cytoprotective and antioxidant activities for male sexual organs and sperm after the administration of polyphenolic-rich plants [82,83,84,85]. Therefore, our study focused specifically on oleanane saponins, lupane triterpenes, and phenolic diterpenes, and to elucidate the mechanism of how they affect spermatogenesis, while polyphenolics were outside the scope of this study.

The classes of compounds which drew our attention were reported to have apoptotic action on different cancer cell lines via up-regulation of caspase-3 or down-regulation of β-catenin, concerning phenolic diterpenes [86,87,88,89], betulinic acid and its natural derivatives [90,91], and oleanane saponins [92,93,94,95]. In particular, sasanguasaponin (oleanane saponin) isolated from defatted seeds of *Camellia oleifera* was reported to induce spermatogenic cell apoptosis in vivo via a caspase-3 activation reaction [72], and *Tripterygium wilfordii* extract, a male contraceptive diterpene-rich plant that demonstrated an apoptosis action on male rat germ cells and induced a modification for epigenetic histone [96]. Therefore, these metabolites could probably induce apoptosis in testes tissues. Our study was directed towards an immunohistochemical assay for the three most bioactive extracts (RO, AL, and AA) by measuring the degree of reaction between each extract and the two proteins (β-catenin and caspase-3). Their expression plays an essential role in causing apoptosis, particularly in testes cells (β-catenin), and generally in other organs cells including testes (caspase-3; Figure 6 and Figure 7). Clearly, spermatid-specific deletion of β-catenin resulted in disruption of adherence junctions between Sertoli cells and elongating spermatids, acrosomal defects, increased germ cell apoptosis, and in consequence, reduced sperm count and impaired fertility. In addition, caspase-3 is a member of the cysteine-aspartic acid protease (caspase) family, where its sequential activation is considered as one of important keys in the cellular apoptosis process [97,98].

Morphometric study of testes sections of RO, AL, and AA after application of β-catenin and caspase-3 immunohistochemistry indicated a significant decrease and increase of immunoexpression of both β-catenin and caspase-3, respectively, in all three treatment groups, which confirmed the occurrence of apoptosis in these specimens (Figure 6 and Figure 7). Moreover, high concentration metabolites, as detected via LC-MS, were subjected to modeling studies measuring the extent of binding between the target compounds and caspase enzyme (Table 2, Figure 5) to prove if these compounds are responsible for the incidence of apoptosis in testes tissues. Results suggested that these high concentration constituents may be substantially contributing to a reduction in spermatogenesis, particularly oleanane saponins, due to their relative binding affinities, and binding interactions with the caspase-3 active site had the highest docking score. 

## 4. Material and Methods

### 4.1. Chemicals and Reagents

HPLC grade acetonitrile was purchased from TEDIA Company Inc. 79 (Fairfield, CT, USA). Ultra-pure water was purified by an EPED super purification system (Nanjing, China). Formic acid was obtained from Merck KGaA (Darmstadt, Germany). Methanol (MeOH) used for extraction, formaldehyde, glacial acetic acid, and ethanol used for the preparation of 10% formalin and Davidson’s solution were purchased from El-Nasr Company for Pharmaceuticals and Chemicals, Egypt, and was distilled before use.

### 4.2. Plant Material

Leaves of *Menthae piperitae* (MP) and *Apium graveolens* (AG) and flowers of *Calendula officinalis* (CO) were collected in February 2016 from plants cultivated in the campus of Cairo University, Cairo, Egypt, while seeds of *Lactuca sativa* (LS) and *Anethum graveolens* (ANG), leaves of *Rosmarinus officinalis* (RO) and calyces of *Hibiscus sabdariffa* (HS) were purchased in February 2016 from Harrraz, an official seed trader, Cairo, Egypt. In addition, aerial parts of *Calotropis procera* (CP) were collected in March 2016 from a desert land, located in the desert Cairo-Ismaili road, Egypt. *Albizia lebbeck* pods (AL) were collected in March 2016 from a field for medicinal plants, El Qanater El Khayreya, Qalyubia, Egypt, and a *Anagallis arvensis* (AA) whole plant was collected in January 2016 from a field found in Banha Capital, Qalyubia, Egypt. Authentication of the plants was established by Professor Abdel-Halim A. Mohammed, Horticultural Research Institute, Department of Flora and Phytotaxonomy Researches, Dokki, Cairo, Egypt. Voucher specimens (2017-BuPD 51–60), respectively, were deposited at the Department of Pharmacognosy, Faculty of Pharmacy, Cairo University, Egypt. 

### 4.3. Plant Extraction

The air-dried, powdered plant parts (1 kg) of previously mentioned plants were extracted by maceration with 80% MeOH at room temperature, and then concentrated under reduced pressure using a rotary evaporator (IKA, Königswinter, Germany) to a syrupy consistency. The concentrated methanolic extract yields were 60, 105, 85, 230, 75, 100, 80, 200, 150, and 120 g of AL, CP, HS, LS, AG, AA, ANG, CO, RO, and MP, respectively. All the resulting dried extracts were kept at 4 °C for the biological and metabolomic investigations. 

### 4.4. Oral Acute Toxicity Study

This study for ten tested samples, previously mentioned, was performed to determine their safe doses [99]. For each extract, 24 adult male rats (Sprague–Dawley) of about 150 ± 20 g were starved overnight and randomly divided into 4 groups (6 rats each). Rats were kept overnight fasting then subjected to a daily sole dose of 1000, 2000, 4000, and 5000 mg/kg b.w of each examined extract. The observation of animals individually performed after 24 and 72 h with a daily observation. After this period, the LD_50_ was calculated, and the pharmacological evaluation of the extracts was carried out at a fixed daily dose of 10% of LD_50_. As shown in Appendix A, all previously reported toxicity studies and LD_50_ of the investigated extracts or their alternatives had a wide therapeutic index and safety margins, and no adverse effects or toxicities were produced except in very high doses and/or chronic administration of few number of evaluated plants. These findings were considered to be in high agreement with our oral acute toxicity study of chosen plant extracts, and thus suggesting that using of fixed daily doses of 10% of LD_50_ for consecutive 60 days could not produce any mortality or serious toxicity of different organs. 

### 4.5. Animals Grouping, Modelling and Drugs Administration

Seventy-seven adult male albino rats (150–200 g each), in agreement with guidelines of the care and use for laboratory animals of the National Institutes of Health [100], was approved by the Research Ethics Committee for Animal Experimentation, Department of Pharmacology and Toxicology, Faculty of Pharmacy, Minia University, Egypt (project code No. 2018:022). Rats were housed and then bred under systematized conditions in the pre-clinical animal house. Rats were kept in stainless steel cages (seven per cage), fed a suitable diet, and allowed unlimited access to drinking water. They were acclimatized to the environment for one week before the commencement of the experiment. All conditions were also made to diminish animal suffering. 

The 77 male rats were divided into 11 groups, each group consisting of seven animals. Each individual rat in each group between the ten treatment groups was orally received 10% of LD_50_ daily dose for 60 days of its specific extract for its definite group. This daily oral dose was 500 mg/kg, b.w for all ten treatment groups except AL and AA groups, of which the dose was 200 mg/kg, b.w, and the control group received the vehicle (normal saline, 2 mL/kg, b.w). At day 60, rats were anaesthetized by thiopental injection (45 mg/kg b.w), then the blood samples were collected using the retino-orbital sinus, centrifuged for 15 min (3000 rpm/min) to obtain the serum that stored at −80 °C for estimation of LH, FSH and testosterone levels. An incision was made longitudinally in the scrotum by the way that both testes were exposed. Cauda epididymis was chosen for the collection of semen samples used in semen analysis. The testes, prostate and seminal vesicles were dissected and weighed. Testes were preserved in Davidson’s solution, but seminal vesicles were preserved in 10% neutral formalin solution until histopathological examination processing.

### 4.6. Hormonal Assay

Blood samples were collected in the fasting state between 07:00 and 10:00 h. Serum LH, total testosterone, and FSH were quantitatively determined by commercial electrochemiluminescence immunoassay methods (Elecsys 2010, Roche Diagnostics, Mannheim, Germany). For all parameters, the interassay and intraassay coefficients of variation were <10% and 8%, respectively.

### 4.7. Semen Analysis 

This analysis in epididymis was performed by the cutting of cauda epididymis surgically by using surgical blades, then squeezed into a sterile clean watch glass to obtain semen content. The content was diluted with 0.09% NaCl isotonic solution and mixed for estimation of the sperm progressive motility and density, as described by [101]. 

### 4.8. Histological Analysis

Sexual organs specimens were fixed in the previously mentioned solutions. They were dehydrated in an increasing gradient of ethanol, cleared in xylene, and embedded in paraffin. Serial sections of 5–7 μm thickness were cut and subjected to H&E staining [102].

### 4.9. Metabolomics and PCA Analyses

The crude hydromethanolic extracts of ten plants, previously mentioned above, were subjected to metabolomic analysis using LC-HR-ESI-MS analytical techniques [103]. Briefly, the total extract of each plant (1 mg/mL in MeOH) was uploaded on an Accela HPLC (Thermo Fisher Scientific, Bremen, Germany) combined with Accela UV–vis and Exactive (Orbitrap) mass spectrometer from Thermo Fisher Scientific (Bremen, Germany). The mobile phase composed of HPLC grade water (A) and acetonitrile (B) with 0.1% formic acid in each solvent. The gradient elution started at a flow rate of 300 μL/min with 10% B linearly increased to 100% B within 30 min and remained isocratic for the next 5 min before linearly decreasing back to 10% B for the following 1 min. Afterwards, the mobile phase was equilibrated for 9 min before the next injection. The mass range was set from *m/z* (mass-to-charge ratio) 100–2000 for ESI-MS using in-source CID (collision-induced dissociation) mechanism and *m/z* 50–1000 for MS/MS using untargeted HCD (high energy collision dissociation). The raw data were imported to MZmine 2.12, a framework for the disparate analysis of mass spectrometry data. Deconvolution of chromatogram was then performed followed by deisotoping of peaks. The retention time normalizer was applied for chromatographic alignment and gap-filling. For the combination of negative and positive ionization mode data files that were generated by MZmine, Excel macros were used. Peaks produced from the samples were then extracted. The Excel macro was used for dereplicating each *m/z* ion peak with constituents in the customized database (using RT and *m/z* threshold of ± 5 ppm), which provided the putative identities of all metabolomes in the total extract(s) in details. The macro was then applied for detection the top 20 features (arranged ascendingly by peak intensity) and the corresponding putative identities by producing a list for the extract. Therefore, diverse constituents were detected in selected extracts by rapprochement with some in-house and online databases. 

The resultant data matrices were introduced to the Extended Statistical tool EZinfo v2.0 software (Umetrics AB, Umeå, Sweden) for multivariate analysis. The data were scaled to unit variance for principal component analysis (PCA) to give an overview of the repeatability of tested samples. The samples with high repeatability should cluster together in the score plot of PCA. 

### 4.10. Modelling Study

Docking calculations were carried out using the molecular operating environment (MOE). The structure of the compounds was built by Chemdraw Ultra 8.0. Docking calculations were carried out on a caspase-3 complex (PDB code 1GFW, http://www.rcsb.org/pdb/home/home.do). A docking simulation was performed on the test compounds with the following protocol. (1) Enzyme structures were checked for missing atoms, bonds, and contacts. (2) Hydrogen atoms were added to the enzyme structure. (3) The ligand molecules were constructed using the builder module and were energy-minimized. (4) The active site was generated using the MOEAlpha site finder. (5) Ligands were docked within the caspase-3 active site using MOEDock with simulated annealing utilized as the search protocol and the CHARMm molecular mechanics force field. (6) The lowest energy conformation of the docked ligand complex was selected and subjected to a further energy minimization using the CHARMm force field. 

To determine the accuracy of this docking protocol, the crystallized ligand, was re-docked into the caspase-3 active site. This procedure was repeated three times, and the best ranked solutions of the ligand exhibited RMSD values of 1.89 Å from the position of the crystallized ligand for caspase-3. In general, RMSD values smaller than 2.0 Å indicate that the docking protocol is capable of accurately predicting the binding orientation of the crystallized ligand. Lowest energy aligned conformation(s) were identified. This protocol was thus deemed to be suitable for the docking of the test compounds into the active site model of the caspase-3 complex.

### 4.11. Immunohistochemical Assay 

Testis specimens were fixed in Davidson’s solution. They were dehydrated in an increasing gradient of ethanol, cleared in xylene, and embedded in paraffin. Serial sections of 5–7 μm thickness were cut and subjected to immunohistochemical staining, as follows: (a) 1-β-catenin (it is a ready-to-use rabbit polyclonal antibody; Santa Cruz Biotechnology, catalogue number sc-7199); and (b) 2-anti-caspase-3 antibody to demonstrate apoptosis (it is a ready-to-use rabbit polyclonal antibody; NEO markers, Thermo Scientific Laboratories, USA, catalogue number RB-1197-R7) [104].

#### 4.11.1. Immunohistochemical Procedure

Sections were incubated in H_2_O_2_ for 15 min then rinsed in PBS (Sigma-Aldrich, Saint Louis, MO, USA). They were boiled in 10 Mm citrate buffer pH 6 (Lab Vision Corporation Laboratories, catalog no AP 9003) for 10 min for antigen retrieval and left to cool in room temperature, then rinsed in PBS. To eliminate non-specific background, sections were incubated with two drops of Ultra V Block for 5 min and then incubated for 60 min with two drops of the primary antibody (this was omitted in the negative control). Afterwards, slides were incubated for 10 min with two drops of biotinylated goat anti-polyvalent secondary antibody at room temperature, then rinsed well with PBS. Followed by incubation with two drops of streptavidin-peroxidase for 10 min at room temperature then washed in PBS. One drop of DAB Plus chromogen was mixed with 2 mL of DAB Plus substrate, then applied to the slides and incubated for 10 min at room temperature, then rinsed well with distilled water. Slides were then counterstained with Mayer’s hematoxylin (Lab Vision Corporation Laboratories, cat no TA-060-MH), dehydrated, and mounted.

#### 4.11.2. Morphometric Study

Data were obtained using a Leica Qwin 500 C image analyzer computer system (Kasr Alaini, Cairo University, Cairo, Egypt). All measurements were done using ×400 magnification and within 10 nonoverlapping fields for each specimen. The following were measured: mean height of spermatogenic epithelium, and area% for β-catenin and caspase-3 immunopositive cells.

### 4.12. Rule of Five and Veber’s Oral Bioavailability Rule of High Feature Compounds

The four Lipinski’s properties molecular weight (MW), log P, hydrogen bond acceptor (HBA), and hydrogen bond donor (HBD) as well as two additional descriptors topological polar surface area (tPSA) and numbers of rotation bonds of compounds (1–15) were calculated by Instant JChem version 17.10.0 from ChemAxon Ltd (Budapest, Hungary) of the molecules and projected onto a drug-like cut-off threshold of Lipinski’s rules and Veber’s oral bioavailability rule (Figure 8). Lipinski’s rule of five, which considers orally active compounds and defines four simple physicochemical parameter ranges (MW ≤ 500, logP ≤ 5, HBD ≤ 5, and HBA ≤ 10) was combined with Veber’s oral bioavailability rule, which includes two additional parameter ranges (tPSA ≤ 140 Å, number of rotatable bonds ≤10) [105,106].

### 4.13. Statistical Analysis

All data were expressed as the mean ± standard error (S.E.) of 7 rats per experimental group. Statistical analysis was performed using a one-way ANOVA test followed by the Student–Newman–Keuls multiple comparisons test by the aid of *Graphpad prism5* and *Graphpad instant2* computer software (San Diego, CA, USA), with values of *p* < 0.05 considered statistically significant.

## 5. Conclusions

The present work reveals the significant effects of hydromethanolic extracts of *A. lebbck* pods and *A. arvensis* and *R. officinalis* leaves on the testicular structure. In addition, metabolomic analysis of these three plants displayed their ability to biosynthesize and accumulate various secondary metabolites, predominantly oleanane saponins in both *A. lebbck* pods and *A. arvensis* and lupane tritepenes and phenolic diterpenes in *R. officinalis* leaves, which largely suggests their contribution to the adverse action on the male reproductive system (antispermatogenesis) through their apoptotic effects on testes tissue mediated by influencing on β-catenin and caspase-3 proteins. These findings might help broaden the application of these metabolites in future phytotherapy to decrease the incidence of pregnancy without using female contraceptives. Future investigation of more molecular aspects and cellular mechanisms of male contraceptive potential of these potent plants and their suggested active/major metabolites is therefore recommended in the near future, especially betulinic acid and its derivatives, which were seen to only violate the logP rule, and thus may be considered as promising drug candidates. Therefore, more studies concerning the safety, efficacy, and reversibility of these suggested drug candidates are required.

## Figures and Tables

**Figure 1 metabolites-10-00031-f001:**
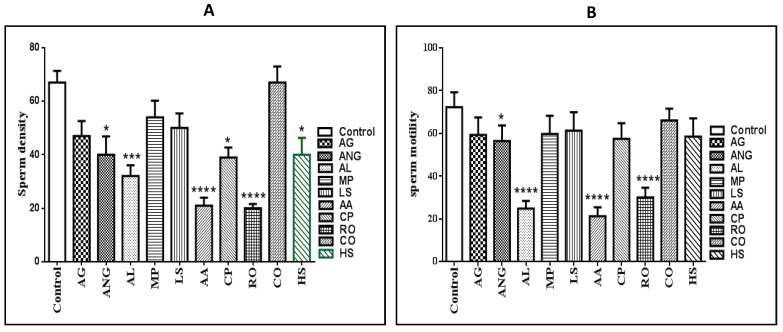
(**A**) Mean epididymal sperm number (×10^6^ cells/mL) of rats exposed to different plant extracts. (**B**) Mean percentage of epididymal sperm motility of rats exposed to different plant extracts. AG: *Apium graveolens*, ANG: *Anethum graveolens*, AL: *Albizzia lebbeck*, MP: *Menthae piperitae*, LS: *Lactuca sativa*, AA: *Anagallis arvensis*, CP: *Calotropis procera*, RO: *Rosmarinus officinalis*, CO: *Calendula officinalis,* and HS: *Hibiscus sabdariffa*. Values are expressed as mean ± S.E.M. For each group, *N* = 7. * Significantly different from control at *p* < 0.05, *** significantly different from control at *p* < 0.001, and **** significantly different from control at *p* < 0.0001.

**Figure 2 metabolites-10-00031-f002:**
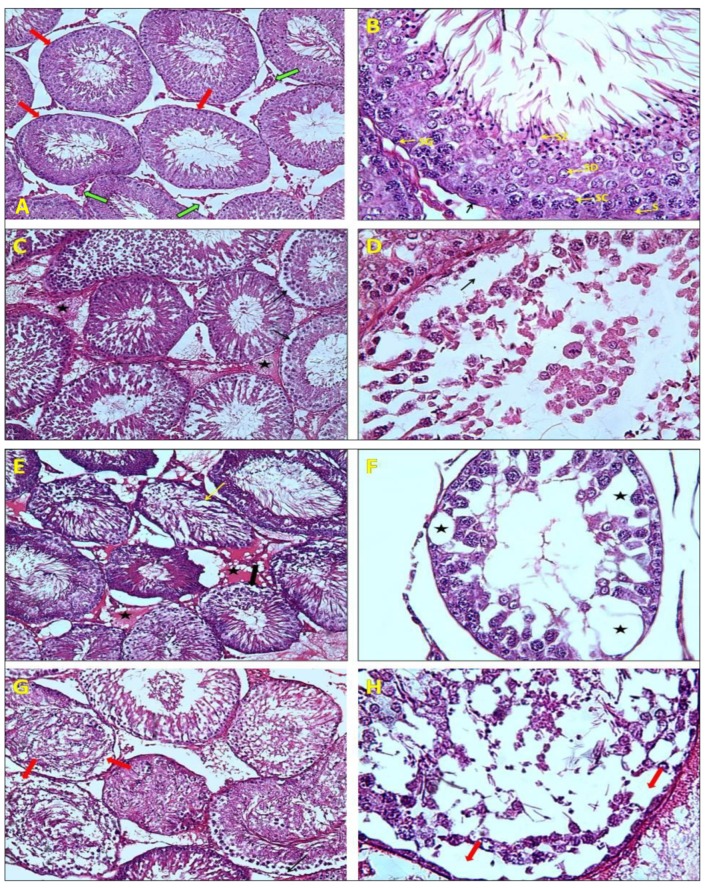
A photomicrograph of a paraffin section in testis. (**A**) Testis of group I (control group) showing densely packed seminiferous tubules (black arrows), separated by an interstitium containing interstitial cells of Leydig and blood vessels (white arrows) (H&E ×200). (**B**) The spermatogenic cells including spermatogonia (SG), primary spermatocytes (SC), spermatids (SD) and spermatozoa (SZ). Sertoli cells (S) had large pale nuclei resting on the basement membrane. The tubules are enclosed by myoid cells (arrow) (H&E ×400). (**C**) Testis of group H (*Rosmarinus officinalis* treated group) showing densely packed seminiferous tubules but with interstitial exudates (Astrix) (H&E ×200). (**D**) Loss of the normal architecture of spermatogenic cells with separation from the basement membrane and appearance of numerous empty spaces (arrow) (H&E ×400). (**E**) Testis of group C (*Albizia lebbeck* pods treated group) showing distorted irregular seminiferous tubules with interstitial exudates (Astrix) and disorganization of spermatogenic epithelium (H&E ×200). (**F**) Marked reduction of germinal epithelium and appearance of a lot of empty spaces (Astrix) (H&E ×400). (**G**) Testis of group F (*Anagallis arvensis* treated group) showing seminiferous tubules with marked detachment (arrows) and disorganization of spermatogenic epithelium (H&E ×200). (**H**) Marked reduction, detachment (arrows), displacement and degeneration of spermatogenic cells (H&E ×400).

**Figure 3 metabolites-10-00031-f003:**
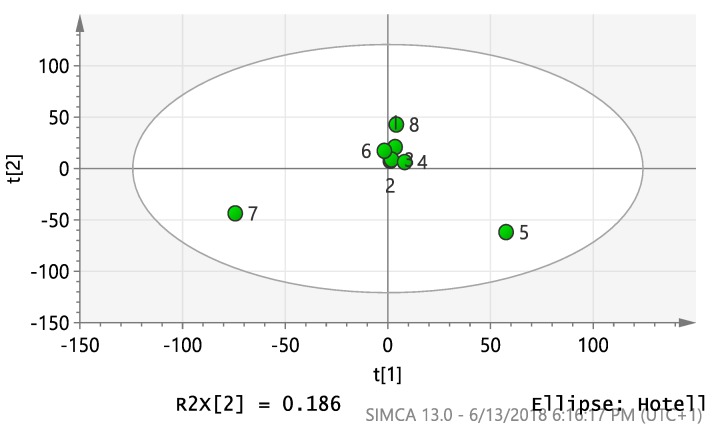
Principal component score plot analysis of ten extracts clustered according to features (*m*/*z* ratios) from mass spectral data (*R*^2^ = 0.4). 1: AG, 2: ANG, 3: MP, 4: RO, 5: AA, 6: CO, 7: AL, 8: CP, 9: LS, and 10: HS.

**Figure 4 metabolites-10-00031-f004:**
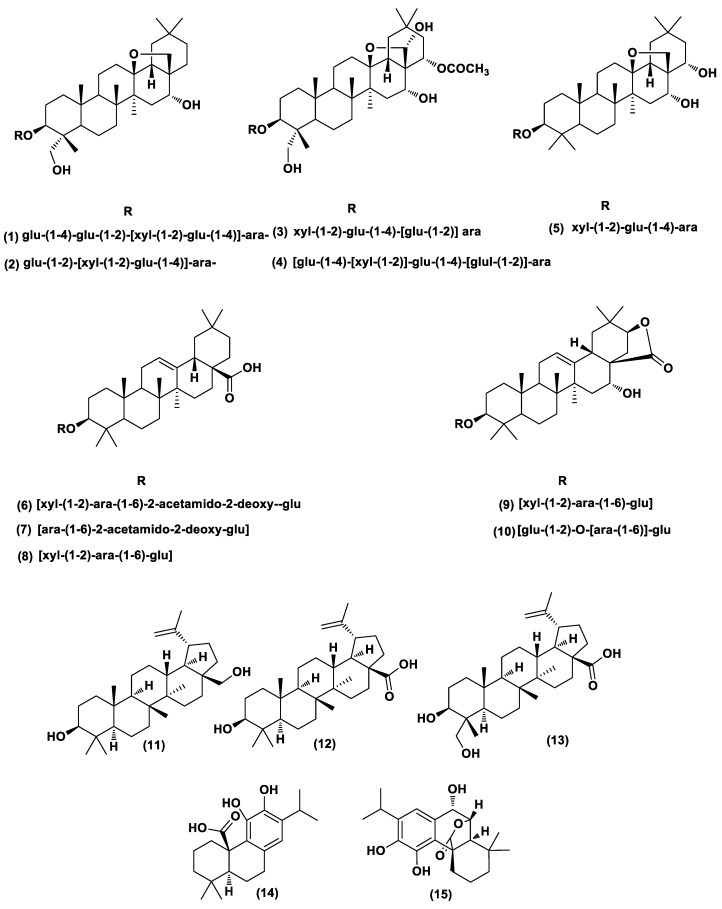
Structures of the dereplicated metabolites from AA (**1**–**5**), AL (**6**–**10**), and RO (**11**–**15**). glu: β-D-glucose, xyl: β-D-xylose, and ara: α-L-arabinose.

**Figure 5 metabolites-10-00031-f005:**
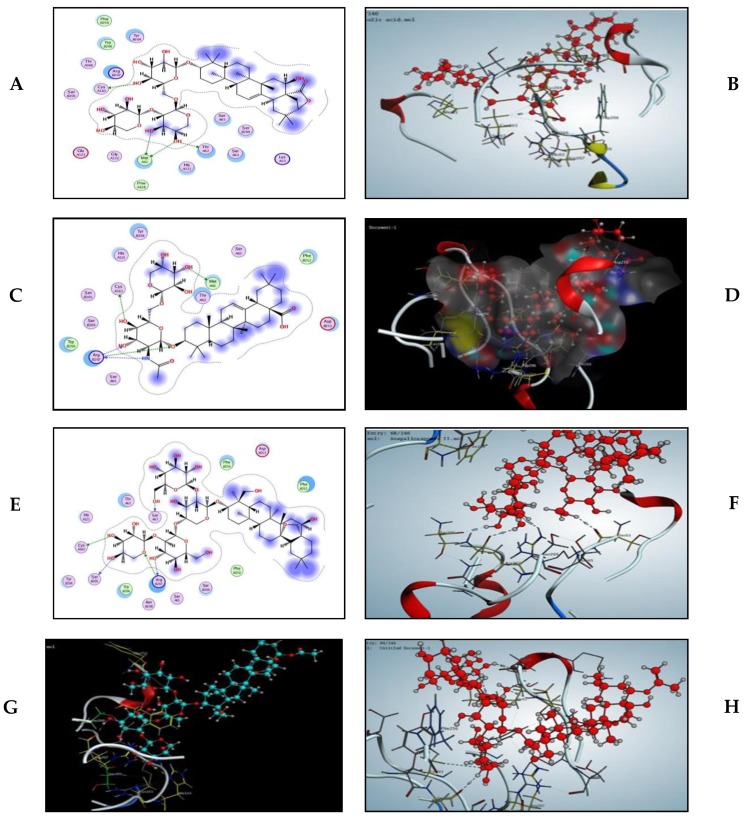
(**A**) The 2D caption of albizeasaponin A binding to the active site of caspase-3. (**B**) Computer modeling of albizeasaponin A binding to Caspase-3 (1GFW). Albizeasaponin A is colored in red. (**C**) The 2D caption of albiziabioside A binding to the active site of caspase-3. (**D**) Binding pattern of albiziabioside A colored by element, ball, and stick into caspase-3 showing 5 hydrogen bond interactions (dotted lines). (**E**) The 2D caption of anagallicin C binding to the active site of caspase-3. (**F**) Computer modeling of anagallicin C binding to Caspase-3 (1GFW). Anagallicin C is colored in red. (**G**) 3D caption of anagallosaponin II colored by element, ball, and stick into caspase-3 showing hydrogen bond interactions (dotted lines), anagallosaponin II colored in blue. (**H**) Binding pattern of anagallosaponin IX colored by element, ball, and stick into caspase-3 showing 6 hydrogen bond interactions (dotted lines).

**Figure 6 metabolites-10-00031-f006:**
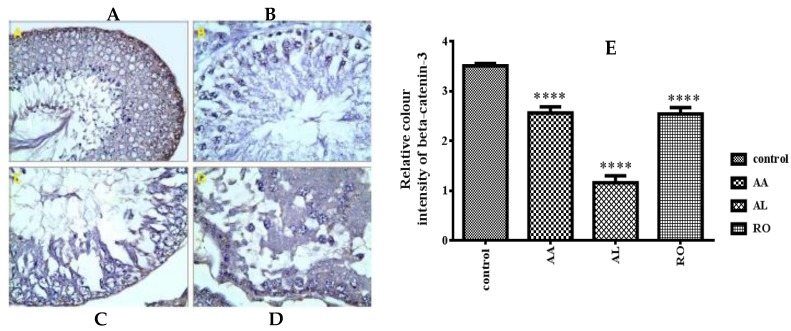
A photomicrograph of a paraffin section in testis (β-catenin ×400). (**A**) Testis of group I (control group) showing strong positive β-catenin reaction in the basal part of germinal epithelium. (**B**,**C**) Testis of the (RO) treated group and the (AL) treated group, respectively, showing weak positive β-catenin reaction in the basal part of germinal epithelium. (**D**) Testis of the (AA) treated group showing moderate positive β-catenin reaction in the basal part of germinal epithelium. (**E**) The mean color area percentage of β-catenin of the RO, AL, and AA groups, measured by the intensity of brown color. **** significantly different from control at *p* < 0.0001.

**Figure 7 metabolites-10-00031-f007:**
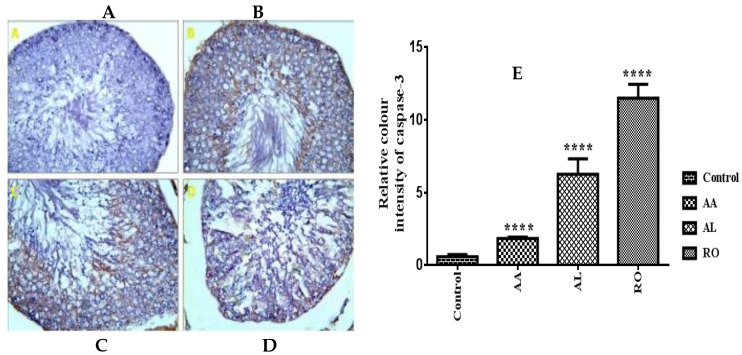
A photomicrograph of a paraffin section in testis (Caspase-3 ×400). (**A**) Testis of group I (control group) showing negative caspase-3 reaction in the cytoplasm of germinal cells. (**B**,**C**) Testis of the RO treated group and the AL treated group, respectively, showing a strong positive caspase-3 reaction in the cytoplasm of germinal cells. (**D**) Testis of the AA treated group showing a moderate positive caspase-3 reaction in the cytoplasm of germinal cells. (**E**) The mean color area percentage of caspase-3 of the RO, AL, and AA groups, measured by intensity of brown color. **** significantly different from control at *p* < 0.0001.

**Figure 8 metabolites-10-00031-f008:**
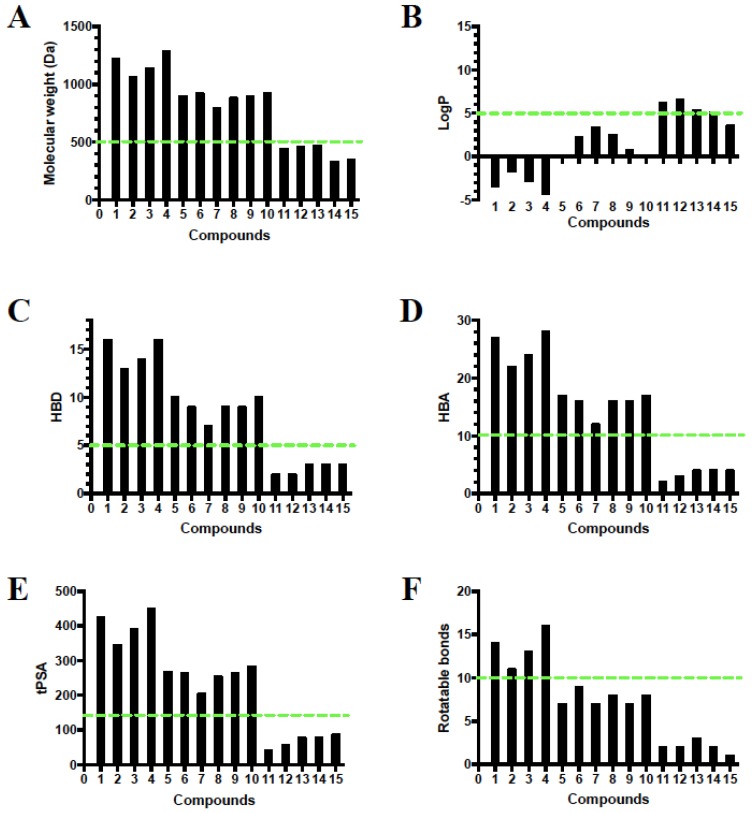
Analysis of physicochemical properties for compounds **1–15** by (**A**) molecular weight, (**B**) log P, (**C**) HBD, (**D**) HBA, (**E**) tPSA, and (**F**) number of rotatable bonds. The green line indicates the maximum desirable value for oral bioavailability defined by Lipinski’s rule of five and Veber’s oral bioavailabilty rule.

**Table 1 metabolites-10-00031-t001:** High features of compounds (ranked by peak intensity) detected in the hydromethanolic extracts of *R. officinalis*, *A. lebbeck*, *A. arvensis* after dereplication of their metabolomes.

No	Accurate *m/z*	Suggested Formula	Ion Adduct	Tentative Detection ^b^	Intensity	Plant Extract
1	1225.6210	C_58_H_96_O_27_	[M + H]^+^	Anagallisin A	8.8 × 10^5^	AA
2	1062.5593	C_52_H_86_O_22_	[M + H]^+^	Anagallisin C	4.4 × 10^7^	AA
3	1136.5636	C_54_H_88_O_25_	[M + H]^+^	Anagallosaponin II	1.2 × 10^7^	AA
4	1282.6160	C_60_H_98_O_29_	[M + H]^+^	Anagallosaponin IX	2.3 × 10^7^	AA
5	900.5069	C_46_H_76_O_17_	[M + H]^+^	Anagallosaponin VI	3.6 × 10^6^	AA
6	923.5234	C_48_H_77_NO_16_	[M + H]^+^	Albiziatrioside A	6.8 × 10^6^	AL
7	792.4895	C_43_H_69_NO_12_	[M + H]^+^	Albiziabioside A	1.1 × 10^4^	AL
8	883.5055	C_46_H_74_O_16_	[M + H]^+^	Pitheduloside C	2.2 × 10^4^	AL
9	897.4844	C_46_H_72_O_17_	[M + H]^+^	Albiziasaponin A	3 × 10^4^	AL
10	927.4945	C_47_H_74_O_18_	[M + H]^+^	Albiziasaponin B	1.9 × 10^4^	AL
11	443.3881	C_30_H_50_O_2_	[M + H]^+^	Betulinol	5.6 × 10^4^	RO
12	457.3677	C_30_H_48_O_3_	[M + H]^+^	Betulinic acid	5.6 × 10^4^	RO
13	473.3622	C_30_H_48_O_4_	[M + H]^+^	Hydroxybetulinic acid	3.4 × 10^4^	RO
14	333.2062	C_20_H_28_O_4_	[M + H]^+^	Carnosic acid	9.6 × 10^6^	RO
15	346.1781	C_20_H_26_O_5_	[M + H]^+^	Rosmanol	1.4 × 10^7^	RO

High-resolution electrospray ionization mass spectrometry (HR-ESI-MS) using XCalibur 3.0 and allowing for M + H/M + Na adduct. ^b^ The suggested compound according to the Dictionary of Natural Products (DNP 23.1, 2015 on DVD) and the Reaxys online database.

**Table 2 metabolites-10-00031-t002:** The binding energy score ranked results of target compounds-caspase-3 complex binding conformations.

Compound Name	Binding Energy Score *	Average Number of Poses per Run
23-OH-betulinic acid	−7.474	10
Albizeasaponin A	−15.227	10
Albiziabioside A	−10.731	10
Albiziasaponin B	−14.591	10
Albiziatrioside A	−8.712	10
Anagallicin A	−11.562	10
Anagallicin C	−12.587	10
Anagallosaponin II	−9.921	10
Anagallosaponin IX	−13.087	10
Anagallosaponin VI	−10.545	10
Betulinic acid	−7.628	10
Betulinol	−9.518	9
Carnosic acid	−9.362	8
Pitheduloside C	−10.143	10
Rosmanol	−9.902	4

* The shown score is the mean of three consecutive runs. The docking method was validated by a successful pose-retrieval docking experiment of the ligand (score: −11.425).

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
