# Peer review of "Testicular Caspase-3 and β-Catenin Regulators Predicted via Comparative Metabolomics and Docking Studies"

_metabolites, 2020, doi:10.3390/metabo10010031_

Round 1

Reviewer 1 Report

Dear authors,

I think that your manuscript has greatly improved.

However, I believe that it must undergo another serious revision regarding proper sentence structure and overall use of the English language, especially in the introduction section (e.g. hazard -> hazardous; also, please standardize past verbal tenses).

Author Response

Reviewer 1 comments:

I think that your manuscript has greatly improved. However, I believe that it must undergo another serious revision regarding proper sentence structure and overall use of the English language, especially in the introduction section (e.g. hazard -> hazardous; also, please standardize past verbal tenses).

The authors would like to thank the reviewer for his valuable comment:

The language of the whole paper was revised by native English speaker and all modifications need were done 

Reviewer 2 Report

The main change in the revised version is rephrasing of the title, however the other weaknesses as mentioned to the orginal version are still present:

-no dose respose experiments

-no tests for adverse effects in other organs than the testis

-no test if the efffects are reversible

Author Response

Reviewer 2 comments:

The main change in the revised version is rephrasing of the title, however the other weaknesses as mentioned to the original version are still present

The authors would like to thank the reviewer for his very deep and interesting comment:

Firstly the change in the manuscript was not the rephrasing only but also the backbone of the manuscript where was highlighted on the comparative study between the matabolomic study and the spermatogenesis changes.

No dose response experiments: The dose response for investigated plant extracts did not exist in our plan and/or overall objectives.

The ongoing work aimed to perform a comparative study between 10 chosen plants that were previously reported to affect male reproductive system in order to (1) to confirm their previously reported results on male reproductive system, then (2) suggesting the mechanistic pathways for the most bioactive extracts using different techniques as shown in the manuscript, but in our future plan we are preparing to carry on dose response study for the most bioactive extracts which cannot be covered in just 10 days

No tests for adverse effects in other organs than the testis

The authors would like to express their thanks for this comment, table S11 was inserted in supplementary data showing all previously reported toxicity studies and LD50 of the investigated ten extracts or their alternatives, it was clear that those extracts had a wide therapeutic index and safety margins, and no adverse effects or toxicities on other body organs rather than male reproductive system were produced except few number of evaluated plants and in restrict cases;

In very high doses (e.g Anagallis arvensis) and our dose is less than this Chronic administration (when using other route of administration rather than oral route e.g. Calendula officinalis & Albizia lebbeck and we only use oral administration Using another type of extract such as aqueous or alcoholic rather than hydroalcoholic eg. Calotropis procera. When using one part of the plant not all our selected part e.g. Calotropis procera (using leaves or flowers alone rather than using the whole aerial parts) and Albizia lebbeck (using seeds alone rather than the whole pods) the LD50 was very low but in our study it was very high No test if the effects are reversible

The authors would like to deeply thank the reviewer comment and this will be included in the future work.

Reviewer 3 Report

The authors addressed my comments and the manuscript can be considered for publication in the current version.

Author Response

Reviewer 3 comments:

The authors addressed my comments and the manuscript can be considered for publication in the current version.

The authors would like to thank the reviewer for his valuable consideration for our article to be suitable for publication in Metabolites Journal.

This manuscript is a resubmission of an earlier submission. The following is a list of the peer review reports and author responses from that submission.

Round 1

Reviewer 1 Report

Dear authors,

The subject addressed in your paper is a very interesting and relevant one nowadays. Please see below my suggestions that I think may improve your manuscript.

Firstly, a major revision regarding proper sentence structure and overall use of the English language is necessary, especially in the first part of the manuscript.

Secondly, some minor revisions should be considered:

In the introduction, when presenting previously reported effects, I believe it should be made clear if they were described for the plant product or extract (e.g. for celery or lettuce); It should be clarified the difference between significant (p<0.05) and highly significant (e.g. p<0.01? p<0.001?); In figure 1, I believe that instead of repeating the phrase “Values …”, it would be better to define the significance limit for “ **** ” and explain the plant products’ acronyms (AG, ANG, etc); Perhaps it would be advisable to rephrase the titles for 1.2 and 1.3 (for example, instead of “Hematoxylin and eosin (H & E) histological results” maybe “Histological analysis” would be more appropriate); In figure 2, some of the arrows are difficult to observe (e.g. the white arrows); further, the group should be mentioned in the explication for each individual picture and the magnification, if it is not the same for all picture (e.g. F seems to be taken at a lower magnification than B, D and H); for some pictures (e.g. G and H – yellow arrows), these isn’t a clear explanation of what do the arrows indicate;

Reviewer 2 Report

The aim of this study was to develop a male contraceptive. Therefore male rat were treated with extracts from ten different plants, respectively, and the sperm cells and testis morphology was studied. No appropriate dose -response treatment was performed, adverse effects on other tissues were neglected, and no reproduction studies were approached: are the decreased sperm parameters indicative for reduced fertilization rates?, are the effects reversible?

In the current state, the data are preliminary and not suitable for publication.

Reviewer 3 Report

The authors investigate the activity of ten plants for male contraception in rats, finding that three plants show significant effects on male fertility. The authors perform untargeted metabolomics of the active plants to screen for active compounds correlating with the observed effects. 

This study is an interesting example of how metabolomics can be leveraged for the investigation of known and unknown active metabolites in plants. 

The manuscript is well written and the introduction provides a broad overview on the topic. However the authors should correct the improper use of the word metabolite "identification" over the manuscript and should better explain which metabolites where unequivocally identified according to the metabolomics standards of identification currently in force (level of metabolite ID..). If the authors used MS/MS spectra, the analysis of these should be reported in the supporting information together with the related data. 

line 166: please clarify the sentence "the mass resolution was....isobaric compounds". This does not experimentally apply to all types of isobaric compounds...

line 168: these features are not identified here, rather "detected". Also, correct this point in tables and over the whole manuscript. 

The authors use the word "dereplication" very often in the manuscript: we suggest to clarify what they mean with this word soon in the manuscript. 

 Minor comments:

line 56-57: please rephrase for clarity

line 312: correct "will focused"